# TREE CROSS ATTENTION

**Leo Feng**
Mila – Université de Montréal & Borealis AI
`leo.feng@mila.quebec`

**Frederick Tung**
Borealis AI
`frederick.tung@borealisai.com`

**Hossein Hajimirsadeghi**
Borealis AI
`hossein.hajimirsadeghi@borealisai.com`

**Yoshua Bengio**
Mila – Université de Montréal
`yoshua.bengio@mila.quebec`

**Mohamed Osama Ahmed**
Borealis AI
`mohamed.o.ahmed@borealisai.com`

## ABSTRACT

Cross Attention is a popular method for retrieving information from a set of context tokens for making predictions. At inference time, for each prediction, Cross Attention scans the full set of $\mathcal{O}(N)$ tokens. In practice, however, often only a small subset of tokens are required for good performance. Methods such as Perceiver IO are cheap at inference as they distill the information to a smaller-sized set of latent tokens $L < N$ on which cross attention is then applied, resulting in only $\mathcal{O}(L)$ complexity. However, in practice, as the number of input tokens and the amount of information to distill increases, the number of latent tokens needed also increases significantly. In this work, we propose Tree Cross Attention (TCA) - a module based on Cross Attention that only retrieves information from a logarithmic $\mathcal{O}(\log(N))$ number of tokens for performing inference. TCA organizes the data in a tree structure and performs a tree search at inference time to retrieve the relevant tokens for prediction. Leveraging TCA, we introduce ReTreever, a flexible architecture for token-efficient inference. We show empirically that Tree Cross Attention (TCA) performs comparable to Cross Attention across various classification and uncertainty regression tasks while being significantly more token-efficient. Furthermore, we compare ReTreever against Perceiver IO, showing significant gains while using the same number of tokens for inference.

## 1 INTRODUCTION

With the rapid growth in applications of machine learning, an important objective is to make inference efficient both in terms of compute and memory. NVIDIA (Leopold, 2019) and Amazon (Barr, 2019) estimate that 80–90% of the ML workload is from performing inference. Furthermore, with the rapid growth in low-memory/compute domains (e.g. IoT devices) and the popularity of attention mechanisms in recent years, there is a strong incentive to design more efficient attention mechanisms for performing inference.

Cross Attention (CA) is a popular method at inference time for retrieving relevant information from a set of context tokens. CA scales linearly with the number of context tokens $\mathcal{O}(N)$. However, in reality many of the context tokens are not needed, making CA unnecessarily expensive in practice. General-purpose architectures such as Perceiver IO (Jaegle et al., 2021) perform inference cheaply by first distilling the contextual information down into a smaller fixed-sized set of latent tokens $L < N$. When performing inference, information is instead retrieved from the fixed-size set of latent tokens $\mathcal{O}(L)$. Methods that achieve efficient inference via distillation are problematic since (1) problems with a high intrinsic dimensionality naturally require a large number of latents, and (2) the number of latents (capacity of the inference model) is a hyperparameter that requires specifying before training. However, in many practical problems, the required model's capacity may not be

known beforehand. For example, in settings where the amount of data increases overtime (e.g., Bayesian Optimization, Contextual Bandits, Active Learning, etc...), the number of latents needed in the beginning and after many data acquisition steps can be vastly different.

In this work, we propose (1) Tree Cross Attention (TCA), a replacement for Cross Attention that performs retrieval, scaling logarithmically $\mathcal{O}(\log(N))$ with the number of tokens. TCA organizes the tokens into a tree structure. From this, it then performs retrieval via a tree search, starting from the root. As a result, TCA selectively chooses the information to retrieve from the tree depending on a query feature vector. TCA leverages Reinforcement Learning (RL) to learn good representations for the internal nodes of the tree. Building on TCA, we also propose (2) ReTreever, a flexible architecture that achieves token-efficient inference.

In our experiments, we show (1) TCA achieves results competitive to that of Cross Attention while only requiring a logarithmic number of tokens, (2) ReTreever outperforms Perceiver IO on various classification and uncertainty estimation tasks while using the same number of tokens, (3) ReTreever's optimization objective can leverage non-differentiable objectives such as classification accuracy, and (4) TCA's memory usage scales logarithmically with the number of tokens unlike Cross Attention which scales linearly.

## 2 BACKGROUND

### 2.1 ATTENTION

Attention retrieves information from a context set $X_c$ as follows:

$$\text{Attention}(Q, K, V) = \text{softmax}(\frac{QK^T}{\sqrt{d}})V$$

where $Q$ is the query matrix, $K$ is the key matrix, $V$ is the value matrix, and $d$ is the dimension of the key and query vectors. In this work, we focus on Cross Attention where the objective is to retrieve information from the set of context tokens $X_c$ given query feature vectors, i.e., $K$ and $V$ are embeddings of the context tokens $X_c$ while $Q$ is the embeddings of a batch of query feature vectors. In contrast, in Self Attention, $K$, $V$, and $Q$ are embeddings of the same set of token $X_c$ and the objective is to compute higher-order information for downstream calculations.

### 2.1.1 PERCEIVER IO

Perceiver IO (Jaegle et al., 2021) is a general attention-based neural network architecture applicable to various tasks. Perceiver IO is composed of a stacked iterative attention encoder ($\mathbb{R}^{N \times D} \to \mathbb{R}^{L \times D}$) and a Cross Attention module where $N$ is the number of context tokens and $L$ is a hyperparameter. Perceiver IO's encoder aims to compress the information of the context tokens to a smaller constant ($L$) number of latent tokens (Latents) whose initialization is learned. More specifically, each of the encoder's stacked iterative attention blocks work as follows:

$$\text{Latents} \leftarrow \text{SelfAttention}(\text{CrossAttention}(\text{Latents}, \mathcal{D}_C))$$

When performing inference, Perceiver IO simply retrieves information from the set of latents to make predictions via cross attention. As a result, all the necessary information needed for performing inference must be compressed into its small fixed number of latents. However, this is not practical for problems with high intrinsic dimensionality.

### 2.2 REINFORCEMENT LEARNING

Markov Decision Processes (MDPs) are defined as a tuple $M = (\mathcal{S}, \mathcal{A}, T, R, \rho_0, \gamma, H)$, where $\mathcal{S}$ and $\mathcal{A}$ denote the state and action spaces respectively, $T(s_{t+1}|s_t, a)$ the transition dynamics, $R(r_{t+1}|s_t, a_t, s_{t+1})$ the reward function, $\rho_0$ the initial state distribution, $\gamma \in (0, 1]$ the discount factor, and $H$ the horizon. In the standard Reinforcement Learning setting, the objective is to optimize a policy $\pi(a|s)$ that maximizes the expected discounted return $E_{\pi, T, \rho_0}[\sum_{t=0}^{H-1} \gamma_t R(r_{t+1}|s_t, a_t, s_{t+1})]$.

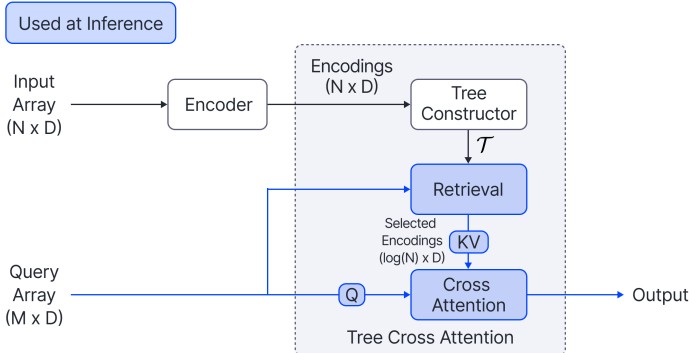

Figure 1: **Architecture Diagram of ReTreever.** Input Array comprises a set of $N$ context tokens which are fed through an encoder to compute a set of context encodings. Query Array denotes a batch of $M$ query feature vectors. Tree Cross Attention organizes the encodings and constructs a tree $\mathcal{T}$. At inference time, given a query feature vector, a logarithmic-sized subset of nodes (encodings) is retrieved from the tree $\mathcal{T}$. The query feature vector retrieves information from the subset of encodings via Cross Attention and makes a prediction.

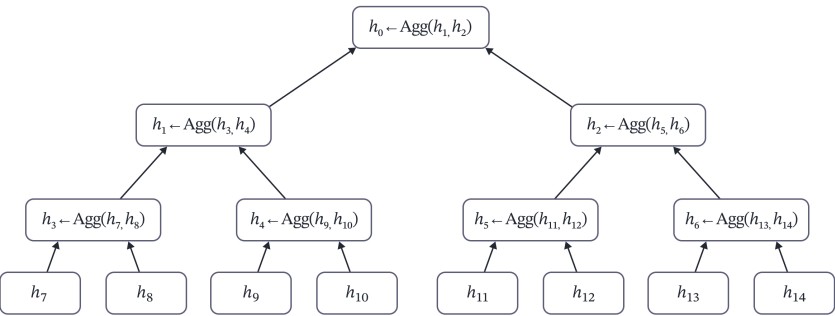

Figure 2: Diagram of the aggregation procedure performed during the Tree Construction phase. The aggregation procedure is performed bottom-up beginning from the parents of the leaves and ending at the root of the tree. The complexity of this procedure is $\mathcal{O}(N)$ but this only needs to be performed once for a set of context tokens. Compared to the cost of performing multiple predictions, the one-time cost of the aggregation process is minor.

## 3  TREE CROSS ATTENTION

We propose Tree Cross Attention (TCA), a token-efficient variant of Cross Attention. Tree Cross Attention is composed of three phases: (1) Tree Construction, (2) Retrieval, and (3) Cross Attention. In the Tree Construction phase ($\mathbb{R}^{N \times D} \to \mathcal{T}$), TCA organizes the context tokens ($\mathbb{R}^{N \times D}$) into a tree structure ($\mathcal{T}$) such that the context tokens are the leaves in the tree. The internal (i.e., non-leaf) nodes of the tree summarise the information in its subtree. Notably, this phase only needs to be performed once for a set of context tokens.

The Retrieval and Cross Attention is performed multiple times at inference time. During the Retrieval phase ($\mathcal{T} \times \mathbb{R}^D \to \mathbb{R}^{\mathcal{O}(\log(N)) \times D}$), the model retrieves a logarithmic-sized selected subset of nodes ($\mathbb{S}$) from the tree using a query feature vector $q \in \mathbb{R}^D$ (or a batch of query feature vectors $\mathbb{R}^{M \times D}$). Afterwards, Cross Attention ($\mathbb{R}^{\mathcal{O}(\log(N)) \times D} \times \mathbb{R}^D \to \mathbb{R}^D$) is performed by retrieving the information from the subset of nodes with the query feature vector. The overall complexity of inference is $\mathcal{O}(\log(N))$ per query feature vector. We detail these phases fully in the below subsections.

After describing the phases of TCA, we introduce ReTreever, a general architecture for token-efficient inference. Figure 1 illustrates the architecture of Tree Cross Attention and ReTreever.

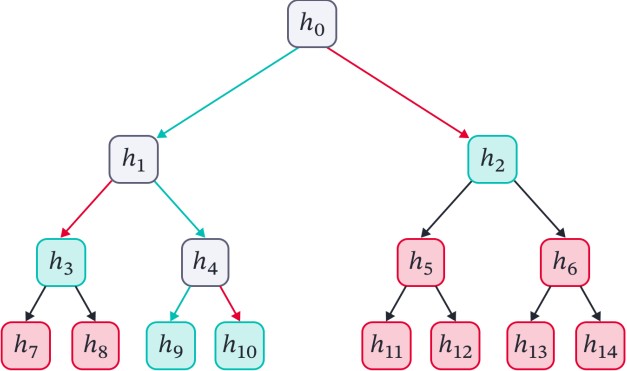

Figure 3: Example result of a Retrieval phase. The policy creates a path from the tree's root to its leaves, selecting a subset of nodes: the terminal leaves in the path and the highest-level unexplored ancestors of the other leaves. The green arrows represent the path (actions) chosen by the policy $\pi$. The red arrows represent the actions rejected by the policy. The green nodes denote the subset of nodes selected, i.e., $\mathbb{S} = \{h_2, h_3, h_9, h_{10}\}$. The grey nodes denote nodes that were explored at some point but not selected. The red nodes denote the nodes that were not explored or selected.

### 3.1 TREE CONSTRUCTION

The tokens are organized in a tree $\mathcal{T}$ such that the leaves of the tree consist of all the tokens. The internal nodes (i.e., non-leaf nodes) summarise the information of the nodes in their subtree. The information stored in a node has two specific desideratas, summarising the information in its subtree needed for performing (1) predictions and (2) retrieval (i.e., finding the specific nodes in its subtree relevant for the query feature vector).

The method of organizing the data in the tree is flexible and can be done either with prior knowledge of the structure of the data, with simple heuristics, or potentially learned. For example, heuristics used to organize the data in traditional tree algorithms can be used to organize the data, e.g., ball-tree (Omohundro, 1989) or k-d tree (Cormen et al., 2006).

After organizing the data in the tree, an aggregator function $\text{Agg} : \mathcal{P}(\mathbb{R}^d) \to \mathbb{R}^d$ (where $\mathcal{P}$ denotes the power set) is used to aggregate the information of the tokens. Starting from the parent nodes of the leaves of the tree, the following aggregation procedure is performed bottom-up until the root of the tree: $h_v = \text{Agg}(\{h_u | u \in \mathbb{C}_v\})$ where $\mathbb{C}_v$ denotes the set of children nodes of a node $v$ and $h_v$ denotes the vector representing the node. Figure 2 illustrates the aggregation process.

In our experiments, we consider a balanced binary tree. To organize the data, we use a k-d tree. During their construction, k-d trees select an axis and split the data evenly according to the median, grouping similar data together. For sequence data such as time series, the data is split according to the x-axis (e.g., time). For images, this means that the pixels are split according to the x or y-axis, i.e., vertically or horizontally. To ensure the tree is perfectly balanced, padding tokens are added such that the values are masked during computation. Notably, these padding nodes are less than the number of tokens. As such, it does not affect the big-O complexity.

### 3.2 RETRIEVAL

To learn informative internal node embeddings for retrieval, we leverage a policy $\pi$ learned via Reinforcement Learning for retrieving a subset of nodes from the tree. Algorithm 1 describes the process of retrieving the set of selected nodes $\mathbb{S}$ from the tree $\mathcal{T}$. Figure 3 illustrates an example of the result of this phase. Figures 7 to 11 in the Appendix illustrate a step-by-step example of the retrieval described in Algorithm 1.

$v$ denotes the node of the tree currently being explored. At the start of the search, it is initialized as $v \leftarrow \text{Root}(\mathcal{T})$. The policy's state is the set of children nodes of the current node being explored $\mathbb{C}_v$ and the query feature vector $q$, i.e., $s = (\mathbb{C}_v, q)$. Starting from the root, the policy samples one of the children nodes, i.e., $v' \sim \pi_\theta(a | \mathbb{C}_v, q)$ where $a \in \mathbb{C}_v$, to further explore for more detailed

information retrieval. The nodes that are not being further explored are added to the set of selected nodes $\mathbb{S} \leftarrow \mathbb{S} \cup (\mathbb{C}_v \backslash v')$. Afterwards, $v$ is updated ($v \leftarrow v'$) with the new node and the process is repeated until $v$ is a leaf node (i.e., the search is complete).

Since the height of a balanced tree is $\mathcal{O}(\log(\mathcal{N}))$, as such, the number of nodes that are explored and retrieved is logarithmic: $||\mathbb{S}|| = \mathcal{O}(\log(N))$.

---

**Algorithm 1** Retrieval

---

**Input:** Tree Structure ($\mathcal{T}$), policy ($\pi_\theta$), and query feature vector $q$
**Output:** Set of selected nodes ($\mathbb{S}$)
  $v \leftarrow \text{Root}(\mathcal{T})$
  $\mathbb{S} \leftarrow \emptyset$
  **while** $v$ is not leaf **do**
    $v' \sim \pi_\theta(a|\mathbb{C}_v, q)$      $\triangleright$ where $a \in \mathbb{C}_v$
    $\mathbb{S} \leftarrow \mathbb{S} \cup (\mathbb{C}_v \backslash v')$    $\triangleright$ $\mathbb{C}_v$ denotes children nodes of $v$
    $v \leftarrow v'$
  **end while**
  $\mathbb{S} \leftarrow \mathbb{S} \cup v$

---

### 3.3 CROSS ATTENTION

The retrieved set of nodes ($\mathbb{S}$) is used as input for Cross Attention alongside the query feature vector $q$. Since $||\mathbb{S}|| = \mathcal{O}(\log(N))$, this results in a Cross Attention with overall logarithmic complexity $\mathcal{O}(\log(N))$ complexity per query feature vector. In contrast, applying Cross Attention to the full set of tokens has a linear complexity $\mathcal{O}(N)$. Notably, the set of nodes $\mathbb{S}$ has a full receptive field of the entire set of tokens, i.e., either a token is part of the selected set of nodes or is a descendant (in the tree) of one of the selected nodes.

### 3.4 ReTreever: RETRIEVAL VIA TREE CROSS ATTENTION

In this section, we propose ReTreever (Figure 1), a general-purpose model that achieves token-efficient inference by leveraging Tree Cross Attention. The architecture is similar in style to Perceiver IO's (Figure 5 in the Appendix).

In the case of Perceiver IO, the model is composed of (1) an iterative attention encoder ($\mathbb{R}^{N \times D} \rightarrow \mathbb{R}^{L \times D}$) which compresses the information into a smaller fixed-sized $L$ set of latent tokens and (2) a Cross Attention module used during inference for performing information retrieval from the set of latent tokens. As a result, Perceiver IO's inference complexity is $\mathcal{O}(L)$.

In contrast, ReTreever is composed of (1) an encoder ($\mathbb{R}^{N \times D} \rightarrow \mathbb{R}^{N \times D}$) and (2) a Tree Cross Attention (TCA) module used during inference for performing information retrieval from a tree structure. Unlike Perceiver IO which compresses information via a specialized encoder to achieve efficient inference, ReTreever's inference is token-efficient irrespective of the encoder, scaling logarithmically $\mathcal{O}(\log(N))$ with the number of tokens. As such, the choice of encoder is flexible and can be, for example, a Transformer Encoder or efficient versions such as Linformer, ChordMixer, etc.

### 3.5 TRAINING OBJECTIVE

The objective ReTreever optimises consists of three components with hyperparameters $\lambda_{RL}$ and $\lambda_{CA}$, denoting the weight of the terms:

$$\mathcal{L}_{ReTreever} = \mathcal{L}_{TCA} + \lambda_{RL}\mathcal{L}_{RL} + \lambda_{CA}\mathcal{L}_{CA}$$

$L_{TCA}$ aims to tackle the first desiderata, i.e., learning node representations in the tree that summarise the relevant information in its subtree for making good predictions.

$$\mathcal{L}_{TCA} = \text{Loss}(\text{CrossAttention}(x, \mathbb{S}), y)$$

$\mathcal{L}_{RL}$ tackles the second desiderata, i.e., learning internal node representations for retrieval (the RL policy $\pi$) within the tree structure[1].

$$\mathcal{L}_{RL} = \sum_{t=0}^{\log(N)-1} [\mathcal{R} \log \pi_\theta(a_t|s_t) + \alpha\mathcal{H}[\pi_\theta(\cdot|s_t)]]$$

---

[1]The described objective is the standard REINFORCE loss (without discounting $\gamma = 1$) using an entropy bonus for a sparse reward environment.

The horizon is the height of the tree $\log(N)$, $\mathcal{H}$ denotes entropy, and $\mathcal{R}$ denotes the reward/objective we want ReTreever to maximise[2]. Typically this reward corresponds to the negative TCA loss $\mathcal{R} = -\mathcal{L}_{TCA}$, e.g., Negative MSE, Log-Likelihood, and Negative Cross Entropy for regression, uncertainty estimation, and classification tasks, respectively. However, crucially, the reward **does not need to be differentiable**. As such, $\mathcal{R}$ can also be an objective we are typically not able to optimize directly via gradient descent, e.g., accuracy for classification tasks ($\mathcal{R} = 1$ for correct classification and $\mathcal{R} = 0$ for incorrect classification).

Furthermore, at inference time, the objective of the policy $\pi$ and the Cross Attention module are intertwined in that the policy aims to select relevant nodes for the Cross Attention. As such, to improve training, the weights are shared between the policy and the Cross Attention module. More specifically, the policy is parameterized as an attention module where the policy's action probabilities are the attention weights of the context (i.e., child nodes $\mathbb{C}_v$) given the query (i.e., $q$).

Lastly, to improve the early stages of training and encourage TCA to learn good node representations, $\mathcal{L}_{CA}$ is included:

$$\mathcal{L}_{CA} = \text{Loss}(\text{CrossAttention}(x, \text{Leaves}(\mathcal{T})), y)$$

## 4 EXPERIMENTS

Our objective in the experiments is: (1) Compare Tree Cross Attention (TCA) with Cross Attention in terms of the number of tokens used and the performance. We answer this in two ways: (i) by comparing Tree Cross Attention and Cross Attention directly on a memory retrieval task and (ii) by comparing our proposed model ReTreever which uses TCA with models which use Cross Attention. (2) Compare the performances of ReTreever and Perceiver while using the same number of tokens. In terms of analyses, we aim to: (3) Highlight how ReTreever and TCA can optimize for non-differentiable objectives using $\mathcal{L}_{RL}$, improving performance. (4) Show the importance of the different loss terms in the training objective. (5) Empirically show the rate at which Cross Attention and TCA memory grow with respect to the number of tokens.

To tackle those objectives, we consider benchmarks for classification (Copy Task, Human Activity) and uncertainty estimation settings (GP Regression and Image Completion). Our focus is on (1) comparing Tree Cross Attention with Cross Attention, and (2) comparing ReTreever with general-purpose models (i) Perceiver IO for the same number of latents and (ii) Transformer (Encoder) + Cross Attention. Full details regarding the hyperparameters are included in the appendix[3].

For completeness, in the appendix, we include the results for many other baselines (including recent state-of-the-art methods) for the problem settings considered in this paper. The results show the baseline Transformer + Cross Attention achieves results competitive with prior state-of-the-art. Specifically, we compared against the following baselines for GP Regression and Image Completion: LBANPs (Feng et al., 2023a), TNPs (Nguyen & Grover, 2022), NPs (Garnelo et al., 2018b), BNPs (Lee et al., 2020), CNPs (Garnelo et al., 2018a), CANPs (Kim et al., 2019), ANPs (Kim et al., 2019), and BANPs (Lee et al., 2020). We compared against the following baselines for Human Activity: SeFT (Horn et al., 2020), RNN-Decay (Che et al., 2018), IP-Nets (Shukla & Marlin, 2019), L-ODE-RNN (Chen et al., 2018), L-ODE-ODE (Rubanova et al., 2019), and mTAND-Enc (Shukla & Marlin, 2021). Additionally, we include results in the Appendix for Perceiver IO with a various number of latent tokens, showing Perceiver IO requires several times more tokens to achieve comparable performance to ReTreever.

### 4.1 COPY TASK

We first verify the ability of Cross Attention and Tree Cross Attention (TCA) to perform retrieval. The models are provided with a sequence of length $N = 2^k$, beginning with a [BOS] (Beginning of Sequence) token, followed by a randomly generated palindrome comprising of $2^k - 2$ digits, ending with a [EOS] (End of Sequence) token. The objective of the task is to predict the second half ($2^{k-1}$) of the sequence given the first half ($2^{k-1}$ tokens) of the sequence as context. To make a correct

---

[2]The only reward $\mathcal{R}$ is given at the end of the episode. As such, the undiscounted return for each timestep is $\mathcal{R}$.

[3]The code is available at https://github.com/BorealisAI/tree-cross-attention

| Method | $N = 256$ | | $N = 512$ | | $N = 1024$ | |
|---|---|---|---|---|---|---|
| | % Tokens | Accuracy | % Tokens | Accuracy | % Tokens | Accuracy |
| Cross Attention | 100.0% | **100.0 ± 0.0** | 100.0% | **100.0 ± 0.0** | 100.0% | **99.9 ± 0.2** |
| Random | — | 8.3 ± 0.0 | — | 8.3 ± 0.0 | — | 8.3 ± 0.0 |
| Perceiver IO | **6.3%** | 15.2 ± 0.0 | **3.5%** | 13.4 ± 0.2 | **2.0%** | 11.6 ± 0.4 |
| TCA | **6.3%** | **100.0 ± 0.0** | **3.5%** | **100.0 ± 0.0** | **2.0%** | **99.6 ± 0.6** |

Table 1: Copy Task Results with accuracy (higher is better) and % tokens (lower is better) metrics.

| Method | % Tokens | RBF | Matern 5/2 |
|---|---|---|---|
| Transformer + Cross Attention | 100% | **1.35 ± 0.02** | **0.91 ± 0.02** |
| Perceiver IO | **14.9%** | 1.06 ± 0.05 | 0.58 ± 0.05 |
| Perceiver IO ($L = 32$) | 68.0% | 1.25 ± 0.04 | 0.78 ± 0.05 |
| ReTreever | **14.9%** | **1.25 ± 0.02** | **0.81 ± 0.02** |

Table 2: 1-D Meta-Regression Experiments with log-likelihood (higher is better) and % tokens (lower is better) metrics.

prediction for index $2^k - i$ (where $0 < i \leq 2^{k-1}$), the model must retrieve information from its input/context sequence at index $i + 1$. The model is evaluated on its accuracy on 3200 randomly generated test sequences. As a point of comparison, we include Random as a baseline, which makes random predictions sampled from the set of digits $(0 - 9)$, [EOS], and [BOS] tokens.

**Results.** We found that both Cross Attention and Tree Cross Attention were able to solve this task perfectly (Table 1). In comparison, TCA requires $\sim 50\times$ fewer tokens than Cross Attention. Furthermore, we found that Perceiver IO's performance for the same number of tokens was dismal ($15.2 \pm 0.0\%$ accuracy at $N = 256$), further dropping in performance as the length of the sequence increased. This result is expected as Perceiver IO aims to compress all the relevant information for any predictions into a small fixed-sized set of latents. However, all the tokens are relevant depending on the queried index. As such, methods which distill information to a lower dimensional space are insufficient. In contrast, TCA performs tree-based memory retrieval, selectively retrieving a small subset of tokens for predictions. As such, TCA is able to solve the task perfectly while using the same number of tokens as Perceiver.

## 4.2 UNCERTAINTY ESTIMATION: GP REGRESSION AND IMAGE COMPLETION

We evaluate ReTreever on popular uncertainty estimation settings used in (Conditional) Neural Processes literature and which have been benchmarked extensively (Table 13 in Appendix) (Garnelo et al., 2018a;b; Kim et al., 2019; Lee et al., 2020; Nguyen & Grover, 2022; Feng et al., 2023a;b).

### 4.2.1 GP REGRESSION

The goal of the GP Regression task is to model an unknown function $f$ given $N$ points. During training, the functions are sampled from a GP prior with an RBF kernel $f_i \sim GP(m, k)$ where $m(x) = 0$ and $k(x, x') = \sigma_f^2 \exp(-\frac{(x-x')^2}{2l^2})$. The hyperparameters of the kernel are randomly sampled according to $l \sim \mathcal{U}[0.6, 1.0]$, $\sigma_f \sim \mathcal{U}[0.1, 1.0]$, $N \sim \mathcal{U}[3, 47)$, and $M \sim \mathcal{U}[3, 50 - N)$. After training, the models are evaluated according to the log-likelihood of functions sampled from GPs with RBF and Matern 5/2 kernels.

**Results.** In Table 2, we see that ReTreever ($1.25 \pm 0.02$) outperforms Perceiver IO ($1.06 \pm 0.05$) by a large margin while using the same number of tokens for inference. To see how many latents Perceiver IO would need to achieve performance comparable to ReTreever, we varied the number of latents (see Appendix Table 13 for full results). We found that Perceiver IO needed $\sim 4.6\times$ the number of tokens to achieve comparable performance.

### 4.2.2 IMAGE COMPLETION

The goal of the Image Completion task is to make predictions for the pixels of an image given a random subset of pixels of an image. The CelebA dataset comprises coloured images of celebrity

| Method | CelebA | | EMNIST | |
|---|---|---|---|---|
| | % Tokens | LogLikelihood | % Tokens | LogLikelihood |
| Transformer + Cross Attention | 100% | **3.88 ± 0.01** | 100% | **1.41 ± 0.00** |
| Perceiver IO | **4.6**% | 3.20 ± 0.01 | **4.6**% | 1.25 ± 0.01 |
| ReTreever | **4.6**% | 3.52 ± 0.01 | **4.6**% | 1.30 ± 0.01 |

Table 3: Image Completion Experiments. The methods are evaluated according to the log-likelihood (higher is better) and % tokens (lower is better) metrics.

| Model | % Tokens | Accuracy |
|---|---|---|
| Transformer + Cross Attention | 100% | **89.1 ± 1.3** |
| Perceiver IO | 14% | 87.6 ± 0.3 |
| ReTreever | 14% | **88.9 ± 0.4** |

Table 4: Human Activity Experiments with accuracy (higher is better) and % tokens (lower is better) metrics.

faces. The images are downsized to size $32 \times 32$. The EMNIST dataset comprises black and white images of handwritten letters with a resolution of $32 \times 32$. We consider 10 classes. Following prior works, the $x$ values of the images are rescaled to be between [-1, 1], the y values to be between [-0.5, 0.5], and the number of pixels is sampled according to $N \sim \mathcal{U}[3, 197]$ and $M \sim \mathcal{U}[3, 200 - N]$.

**Results.** In Table 3, we see that ReTreever outperforms Perceiver IO significantly on both CelebA and EMNIST while using the same number of tokens. Compared with Transformer + Cross Attention, ReTreever uses $\sim 21\times$ fewer tokens, making it significantly more token-efficient.

### 4.3 TIME SERIES: HUMAN ACTIVITY

The human activity dataset consists of 3D positions of the waist, chest, and ankles (12 features) collected from five individuals performing various activities such as walking, lying, and standing. We follow prior works preprocessing steps, constructing a dataset of $6,554$ sequences with 50 time points. The objective of this task is to classify each time point into one of eleven types of activities.

**Results.** Table 4 show similar conclusions as our prior experiments. (1) ReTreever performs comparable to Transformer + Cross Attention, requiring far fewer tokens (86% less), and (2) ReTreever outperforms Perceiver IO significantly while using the same number of tokens.

### 4.4 ANALYSES

**Optimising non-differentiable objectives using $\mathcal{L}_{RL}$.** In classification, typically, accuracy is the metric we truly care about. However, accuracy is a non-differentiable objective, so instead in deep learning, we often use cross entropy so we can optimize the model's parameters via gradient descent. However, in the case of RL, the reward does not need to be differentiable. As such, we compare the performance of ReTreever when optimizing for the accuracy and the negative cross entropy. Table 5 shows that accuracy as the RL reward improves upon using negative cross entropy. This is expected since (1) the metric we evaluate our model on is accuracy and not cross entropy, and (2) accuracy as a reward is simpler compared to negative cross entropy as a reward, making it easier for a policy to optimize. More specifically, a reward using accuracy as the metric is either: $\mathcal{R} = 0$ for an incorrect prediction or $\mathcal{R} = 1$ for a correct prediction. However, a reward based on cross entropy can have vastly different values for incorrect predictions.

| Method | $N = 256$ | | $N = 512$ | | $N = 1024$ | |
|---|---|---|---|---|---|---|
| | % Tokens | Accuracy | % Tokens | Accuracy | % Tokens | Accuracy |
| TCA (Acc) | **6.3**% | **100.0 ± 0.0** | **3.5**% | **100.0 ± 0.0** | **2.0**% | **99.6 ± 0.6** |
| TCA (Neg. CE) | **6.3**% | **99.8 ± 0.3** | **3.5**% | 95.5 ± 3.8 | **2.0**% | 80.3 ± 14.8 |

Table 5: Comparison of Accuracy and Negative Cross Entropy as the reward.

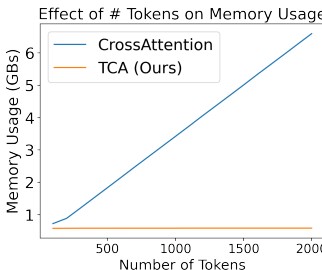 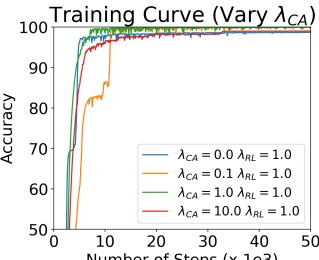 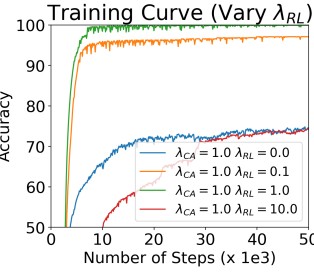

Figure 4: Analyses Plots. (left) Memory usage plot comparing the rate in which memory usage grows at inference time relative to the number of tokens. (middle) Training curve with varying weights ($\lambda_{CA}$) for the Cross Attention loss term. (right) Training curve with varying weights ($\lambda_{RL}$) for the RL retrieval loss term.

**Memory Usage.** Tree Cross Attention's memory usage (Figure 4 (left)) is significantly more efficient than Cross Attention, growing logarithmically in the number of tokens compared to linearly.

**Importance of the different loss terms $\lambda_{CA}$ and $\lambda_{RL}$.** Figure 4 (right) shows that the weight of the RL loss term is crucial to achieving good performance. $\lambda_{RL} = 1.0$ performed the best. If the weight of the term is too small $\lambda_{RL} \in \{0.0, 0.1\}$ or too large $\lambda_{RL} = 10.0$, the model is unable to solve the task. Figure 4 (middle) shows that the weight of the Cross Attention loss term improves the stability of the training, particularly in the early stages. Setting $\lambda_{CA} = 1.0$ performed the best. Setting too small values for $\lambda_{CA} \in \{0.0, 0.1\}$ causes some erraticness in the early stages of training. Setting too large of a value $\lambda_{CA} = 10.0$ slows down training.

## 5 RELATED WORK

There have been a few works which have proposed to leverage a tree-based architecture (Tai et al., 2015; Nguyen et al., 2020; Madaan et al., 2023; Wang et al., 2019) for attention, the closest of which to our work is Treeformer (Madaan et al., 2023). Unlike prior works (including Treeformer) which focused on replacing self-attention in transformers with tree-based attention, in our work, we focus on replacing cross attention with a tree-based cross attention mechanism for efficient memory retrieval for inferences. Furthermore, Treeformer (1) uses a decision tree to find the leaf that closest resembles the query feature vector, (2) TF-A has a partial receptive field comprising only the tokens in the selected leaf, and (3) TF-A has a worst-case linear complexity in the number of tokens per query. In contrast, TCA (1) learns a policy to select a subset of nodes, (2) retrieves a subset of nodes with a full receptive field, and (3) has a guaranteed logarithmic complexity.

As trees are a form of graphs, Tree Cross Attention bears a resemblance to Graph Neural Networks (GNNs). The objectives, however, of GNNs and TCA are different. In GNNs, the objective is to perform edge, node, or graph predictions. However, the goal of TCA is to search the tree for a subset of nodes that is relevant for a query. Furthermore, unlike GNNs which typically consider the tokens to correspond one-to-one with nodes of the graph, TCA only considers the leaves of the tree to be tokens. We refer the reader to surveys on GNNs (Wu et al., 2020; Thomas et al., 2023).

## 6 CONCLUSION

In this work, we proposed Tree Cross Attention (TCA), a variant of Cross Attention that only requires a logarithmic $\mathcal{O}(\log(N))$ number of tokens when performing inferences. By leveraging RL, TCA can optimize non-differentiable objectives such as accuracy. Building on TCA, we introduced ReTreever, a flexible architecture for token-efficient inference. We evaluate across various classification and uncertainty prediction tasks, showing (1) TCA achieves performance comparable to Cross Attention while being significantly more token efficient and (2) ReTreever outperforms Perceiver IO while using the same number of tokens for inference.

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

## A APPENDIX: ILLUSTRATIONS

### A.1 BASELINE ARCHITECTURES

Figures 5 and 6 illustrate the difference in architectures between the main baselines considered in the paper. Transformer + Cross Attention is composed of a Transformer encoder ($\mathbb{R}^{N \times D} \to \mathbb{R}^{N \times D}$) and a Cross Attention module ($\mathbb{R}^{M \times D} \times \mathbb{R}^{N \times D} \to \mathbb{R}^{M \times D}$). Perceiver IO is composed of an iterative attention encoder ($\mathbb{R}^{N \times D} \to \mathbb{R}^{L \times D}$) and a Cross Attention module ($\mathbb{R}^{M \times D} \times \mathbb{R}^{L \times D} \to \mathbb{R}^{M \times D}$). In contrast, ReTreever (Figure 1) is composed of a flexible encoder ($\mathbb{R}^{N \times D} \to \mathbb{R}^{N \times D}$) and a Tree Cross Attention module.

Empirically, we showed that Transformer + Cross Attention achieves good performance. However, its Cross Attention is inefficient due to retrieving from the full set of tokens. Notably, Perceiver IO achieves its inference-time efficiency by performing a compression via an iterative attention encoder. To achieve token-efficient inferences, the number of latents needs to be significantly less than the number of tokens originally, i.e., $L << N$. However, this is not practical in hard problems since setting low values for $L$ results in significant information loss due to the latents being a bottleneck. In contrast, ReTreever is able to perform token-efficient inference while achieving better performance than Perceiver IO for the same number of tokens. ReTreever does this by using Tree Cross Attention to retrieve the necessary tokens, only needing a logarithmic number of tokens $\log(N) << N$, making it efficient regardless of the encoder used. Crucially, this also means that ReTreever is more flexible with its encoder than Perceiver IO, allowing for customizing the kind of encoder depending on the setting.

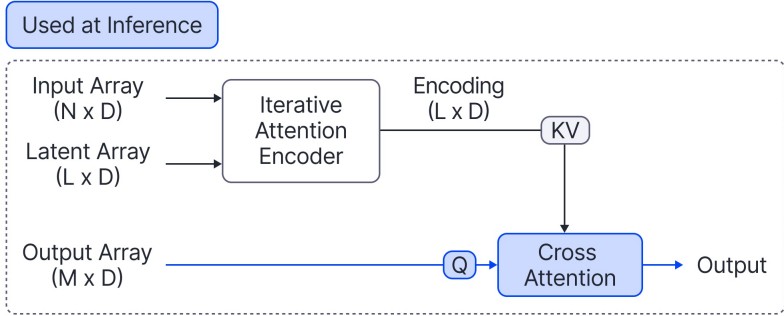

Figure 5: Architecture Diagram of Perceiver IO. The model is composed of an iterative attention encoder ($\mathbb{R}^{N \times D} \to \mathbb{R}^{L \times D}$) and a Cross Attention module ($\mathbb{R}^{M \times D} \times \mathbb{R}^{L \times D} \to \mathbb{R}^{M \times D}$).

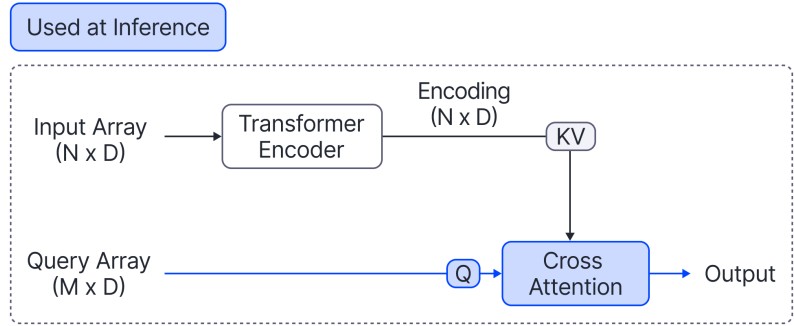

Figure 6: Architecture Diagram of Transformer + Cross Attention. The model is composed of a Transformer encoder ($\mathbb{R}^{N \times D} \to \mathbb{R}^{N \times D}$) and a Cross Attention module ($\mathbb{R}^{M \times D} \times \mathbb{R}^{N \times D} \to \mathbb{R}^{M \times D}$).

### A.2 EXAMPLE RETRIEVAL

In Figures 7 to 11, we illustrate an example of the retrieval process described in Algorithm 1.

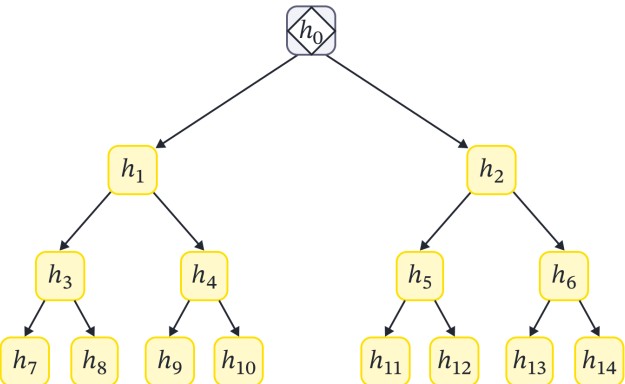

Figure 7: Example Retrieval Step 0 Illustration. The diamond icon on a node indicates the node that will be explored. The retrieval process begins at the root, i.e., $v \leftarrow 0$ where $0$ is the index of the root node. Yellow nodes denote the nodes that have yet to be explored and may be explored in the future. Grey nodes denote the nodes that are or have been explored but have not been added to the selected set $\mathbb{S}$. Red nodes denote the nodes that have not been explored and will not be explored in the future. Green nodes denote the subset of nodes selected, i.e., $\mathbb{S}$.

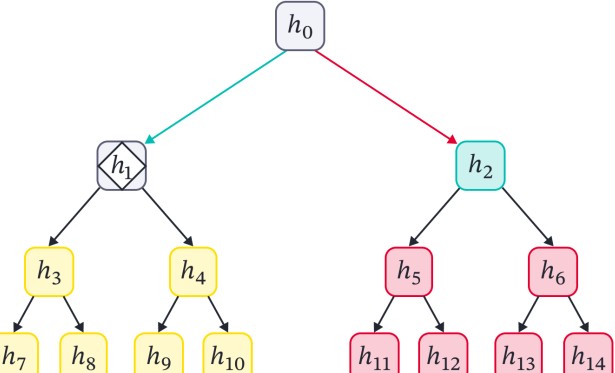

Figure 8: Example Retrieval Step 1 Illustration. Green arrows represent the path (actions) chosen by the policy $\pi$ for a query feature vector. Red arrows represent the actions rejected by the policy. The right child of $v = 0$ was rejected, so it is added to the selected set $\mathbb{S}$. As a consequence, descendant nodes of $v$'s right child will never be explored, so they are coloured red in the diagram. Tentatively, $\mathbb{S} = \{h_2\}$. Because the policy selected the left child of $v = 0$, we thus have $v \leftarrow 1$

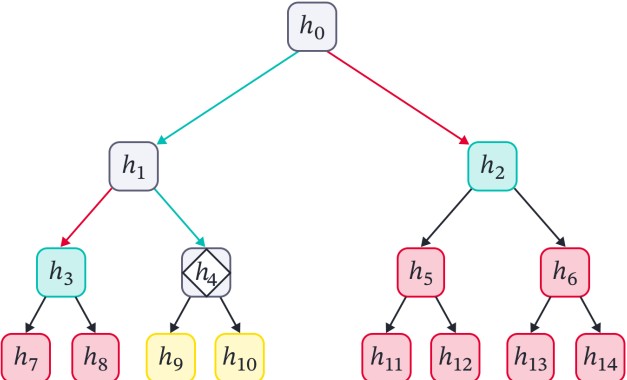

Figure 9: Example Retrieval Step 2 Illustration. The left child of $v = 1$ is rejected this time, so it is added to the selected set $\mathbb{S}$ (green). Tentatively, $\mathbb{S} = \{h_2, h_3\}$. Consequently, the descendant nodes of $v$'s left child will never be explored, so they are coloured red. Because the policy selected the right child of $v = 1$, we thus have $v \leftarrow 4$.

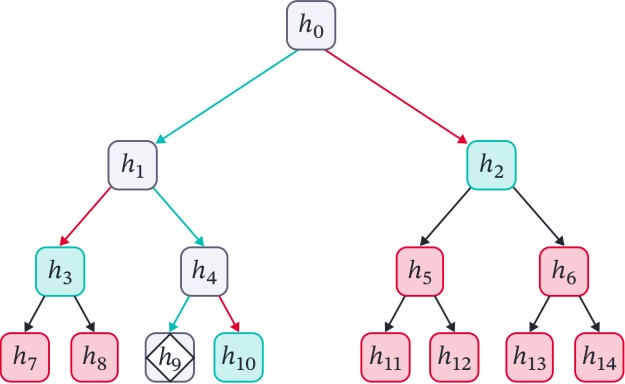

Figure 10: Example Retrieval Step 3 Illustration. The right child of $v = 4$ is rejected this time, so it is added to the selected set $\mathbb{S}$. Tentatively, $\mathbb{S} = \{h_2, h_3, h_{10}\}$. The descendant nodes of $v$'s right child are to be coloured red, but there are none. Because the policy selected the left child of $v = 4$, we thus have $v \leftarrow 9$

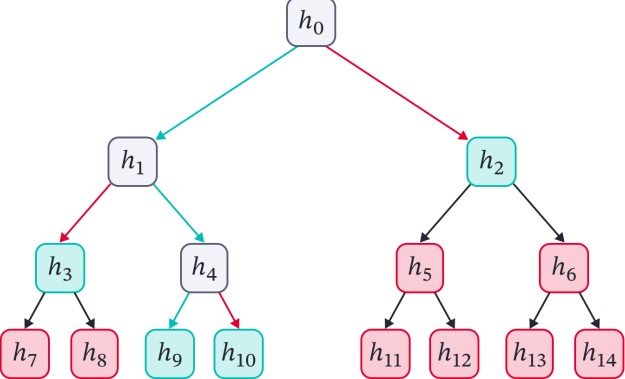

Figure 11: Example Retrieval Step 4 Illustration. $v = 9$ is a leaf, so the tree search has reached its end. At the end of the tree search, we simply add the node to the selected set, resulting in $\mathbb{S} = \{h_2, h_3, h_9, h_{10}\}$.

| Method | % Tokens | CelebA | EMNIST |
|---|---|---|---|
| Transformer + Cross Attention | 100% | $\mathbf{3.88 \pm 0.01}$ | $\mathbf{1.41 \pm 0.00}$ |
| Perceiver IO | **4.6**% | $3.20 \pm 0.01$ | $1.25 \pm 0.00$ |
| ReTreever-Random | **4.6**% | $3.12 \pm 0.02$ | $1.26 \pm 0.01$ |
| ReTreever-KD-$x_0$ | **4.6**% | $3.50 \pm 0.02$ | $\mathbf{1.30 \pm 0.01}$ |
| ReTreever-KD-$x_1$ | **4.6**% | $\mathbf{3.52 \pm 0.01}$ | $1.26 \pm 0.02$ |
| ReTreever-KD-RP | **4.6**% | $3.51 \pm 0.01$ | $1.29 \pm 0.02$ |

Table 6: Tree Design Analyses Experiments. We compare the performance of ReTreever with various heuristics for structuring the tree (1) organized randomly (ReTreever-Random), (2) sorting via the first index $x_0$ of the image data (ReTreever-KD-$x_0$), and (3) sorting via the second index $x_1$ of the image data (ReTreever-KD-$x_1$), and (4) sorting the value after a Random Projection (ReTreever-KD-RP).

| Method | GP Regression | | | Image Completion | |
|---|---|---|---|---|---|
| | % Tokens | RBF | Matern 5/2 | % Tokens | CelebA |
| ReTreever | **14.9**% | $1.25 \pm 0.02$ | $0.81 \pm 0.02$ | **4.6**% | $3.52 \pm 0.01$ |
| ReTreever-Full | 100% | $\mathbf{1.33 \pm 0.02}$ | $\mathbf{0.90 \pm 0.01}$ | 100% | $\mathbf{3.77 \pm 0.01}$ |

Table 7: Comparison of ReTreever with ReTreever-Full which naively selects all the leaves of the tree instead of leveraging the learned policy to select a subset of the nodes.

# B APPENDIX: ADDITIONAL EXPERIMENTS AND ANALYSES

## B.1 ADDITIONAL ANALYSES

### B.1.1 IMPORTANCE OF TREE DESIGN

Unlike sequential data where sorting according to its index is a natural way to organize the data, it is less clear how to organize other data modalities (e.g., pixels) as leaves in a balanced tree. As such, in this experiment (Table 6), we compare the performance of ReTreever with various heuristics for structuring the tree (1) organized randomly (ReTreever-Random), (2) sorting via the first index $x_0$ of the image data (ReTreever-KD-$x_0$), and (3) sorting via the second index $x_1$ of the image data (ReTreever-KD-$x_1$), and (4) sorting the value after a Random Projection (ReTreever-KD-RP)[4]. ReTreever-KD-RP generates a random matrix $w \in \mathbb{R}^{dim(x) \times 1}$ that is fixed during training and orders the context tokens in the tree according to $w^T x$ instead.

In Table 6, we see that ReTreever outperforms Perceiver IO significantly when splitting according to either of the axes. In contrast, organizing the tree randomly causes ReTreever to perform worse than Perceiver IO. There is, however, a minor difference between KD-$x_0$ and KD-$x_1$. We hypothesize that this is in part due to image data being multi-dimensional and not all dimensions are equal.

### B.1.2 RETREEVER-FULL

ReTreever leverages RL to learn good node representations for informative retrieval. However, ReTreever is not limited to only using the learned policy for retrieval at inference time. For example, instead of retrieving a subset of the nodes according to a policy, we can (as a proof of concept) choose to retrieve all the leaves of the tree "ReTreever-Full" (see Table 7).

**Results.** On CelebA and GP Regression, we see that ReTreever-Full achieves performance close to state-of-the-art. For example, on CelebA, ReTreever-Full achieves $3.77 \pm 0.01$, and Transformer + Cross Attention achieves $3.88 \pm 0.01$. On GP, ReTreever-Full achieves $1.33 \pm 0.02$, and Transformer + Cross Attention achieves $1.35 \pm 0.02$. In contrast, ReTreever achieves significantly lower results. The performance gap between ReTreever and ReTreever-Full suggests that the performance of ReTreever can vary depending on the number of nodes (tokens) selected from the tree. As such, an interesting future direction is the design of alternative methods for selecting subsets of nodes.

---

[4]Random Projection is popular in dimensionality reduction literature as it preserves partial information regarding the distances between the points.

| Method | $N = 256$ | | $N = 512$ | | $N = 1024$ | |
|---|---|---|---|---|---|---|
| | % Tokens | Accuracy | % Tokens | Accuracy | % Tokens | Accuracy |
| Cross Attention | 100.0% | $\mathbf{100.0 \pm 0.0}$ | 100.0% | $\mathbf{100.0 \pm 0.0}$ | 100.0% | $\mathbf{99.9 \pm 0.2}$ |
| Random | — | $8.3 \pm 0.0$ | — | $8.3 \pm 0.0$ | — | $8.3 \pm 0.0$ |
| Perceiver IO | $\mathbf{6.3\%}$ | $15.2 \pm 0.0$ | $\mathbf{3.5\%}$ | $13.4 \pm 0.2$ | $\mathbf{2.0\%}$ | $11.6 \pm 0.4$ |
| Perceiver IO | 100.0% | $63.6 \pm 45.4$ | 100.0% | $14.1 \pm 2.3$ | 100.0% | OOM |
| TCA | $\mathbf{6.3\%}$ | $\mathbf{100.0 \pm 0.0}$ | $\mathbf{3.5\%}$ | $\mathbf{100.0 \pm 0.0}$ | $\mathbf{2.0\%}$ | $\mathbf{99.6 \pm 0.6}$ |

Table 8: Copy Task Results with accuracy (higher is better) and % tokens (lower is better) metrics.

| Perceiver IO | Copy Task | | |
|---|---|---|---|
| Num. Latents ($L$) | $N = 128$ | $N = 256$ | $N = 512$ |
| $L = 16$ | $39.83 \pm 40.21$ | $15.16 \pm 0.03$ | $13.41 \pm 0.15$ |
| $L = 32$ | $58.92 \pm 47.44$ | $34.75 \pm 39.14$ | $19.41 \pm 12.07$ |
| $L = 64$ | $79.45 \pm 41.11$ | $27.10 \pm 21.23$ | $14.00 \pm 0.72$ |
| $L = 128$ | $100.00 \pm 0.00$ | $63.50 \pm 39.37$ | $13.23 \pm 0.42$ |

Table 9: Analysis of Perceiver IO's performance relative to the number of latents ($L$) and the sequence length ($N$) on the Copy Task

### B.1.3 PERCEIVER IO'S COPY TASK PERFORMANCE

In Table 8, we include results evaluating Perceiver IO with increased number of latent tokens on varying lengths of the copy task. Specifically, we tested using 100% tokens to evaluate its performance. Perceiver IO's performance degraded sharply as the difficulty of the task increased. Furthermore, due to Perceiver IO's encoder which aims to map the context tokens to a set of latent tokens, we ran out of memory (on a Nvidia P100 GPU (16 GB)) trying to train the model on a sequence length of 1024.

To analyze the reason behind the degradation in performance of Perceiver IO, we evaluated Perceiver IO (Table 9) with a varying number of latent tokens $L \in \{16, 32, 64, 128\}$ and varying sequence lengths $N \in \{128, 256, 512\}$. Generally, we found that performance improved as the number of latents increased. However, the results were relatively unstable. For some runs, Perceiver IO was able to solve the task completely. Other times, Perceiver IO got stuck in lower accuracy local optimas. For longer sequences $N = 512$, increasing the number of latents did not make a difference as the model simply got stuck in poor local optimas.

We hypothesize that the poor performance is due to the incompatibility between the nature of the Copy Task and Perceiver IO's model. The copy task (1) has high intrinsic dimensionality that scales with the number of tokens and (2) does not require computing higher-order information between the tokens. In contrast, Perceiver IO (1) is designed for tasks with lower intrinsic dimensionality by compressing information via its latent tokens and iterative attention encoder, and (2) computes higher-order information via a transformer in the latent space. Naturally, these factors make learning the task difficult for Perceiver IO.

### B.1.4 RUNTIME ANALYSES

**Runtime of Tree Construction and Aggregation.** For 256 tokens, the wall-clock time for constructing a tree with our implementation was $0.076 \pm 0.022$ milliseconds. The wall-clock time for the aggregation is $12.864 \pm 2.792$ milliseconds.

**Runtime of Tree Cross Attention-based models vs. vanilla Cross Attention-based models.** In table 10, we measured the wall-clock time of the modules used during inference time, i.e., ReTreever's Tree Cross Attention module ($\mathbb{R}^{M \times D} \times \mathcal{T} \to \mathbb{R}^{M \times D}$), Transformer+Cross Attention's Cross Attention module ($\mathbb{R}^{M \times D} \times \mathbb{R}^{N \times D} \to \mathbb{R}^{M \times D}$), and Perceiver IO's Cross Attention module ($\mathbb{R}^{M \times D} \times \mathbb{R}^{L \times D} \to \mathbb{R}^{M \times D}$).

We evaluated the modules on the Copy Task for both a GPU and CPU. Although Tree Cross Attention is slower, all methods only require milliseconds to perform inference. Both Tree Cross Attention and Cross Attention learn to solve the task perfectly. However, Tree Cross Attention uses

| Model | % Tokens | Accuracy | CPU time | GPU Time |
|---|---|---|---|---|
| Cross Attention | 100.0% | **100.0 ± 0.0** | 12.05 | 1.61 |
| Tree Cross Attention | **3.5%** | **100.0 ± 0.0** | 19.31 | 9.09 |
| Perceiver IO's CA | **3.5%** | 13.4 ± 0.2 | **10.98** | **1.51** |

Table 10: Comparison of runtime (in milliseconds) for the inference modules of ReTreever, Transformer + Cross Attention, and Perceiver IO, i.e., Tree Cross Attention, Cross Attention, and Perceiver IO's CA respectively. Runtime is reported as the average of 10 runs.

| Model | Tree Height ($H$) | % Tokens | CPU Time (in ms) | GPU Time (in ms) |
|---|---|---|---|---|
| Cross Attention | N/A | 100.0 % | 12.05 | 1.61 |
| TCA ($b_f = 256$) | 1 | 100.0 % | 16.21 | 2.05 |
| TCA ($b_f = 32$) | 2 | 15.2 % | 14.05 | 3.99 |
| TCA ($b_f = 16$) | 2 | 12.1 % | 13.25 | 4.13 |
| TCA ($b_f = 8$) | 3 | 7.0 % | 15.53 | 5.73 |
| TCA ($b_f = 4$) | 3 | 3.9 % | 16.30 | 6.45 |
| TCA ($b_f = 2$) | 8 | 3.5 % | 21.92 | 9.83 |

Table 11: Memory-Runtime trade-off with respect to the tree height ($H$).

only $3.5\%$ of the tokens. As such, Tree Cross Attention tradesoff between the number of tokens and computational time. Perceiver IO, however, fails to solve the task for the same number of tokens.

**The effect of tree design on Memory-Runtime trade-off.** Tree Cross Attention sequentially searches down a tree for the relevant tokens. As such, the factor which primarily affects the overall runtime is the height of tree ($H$) which is dependent on the branching factor $b_f \geq 2$. The relationship between the height of the tree, the branching factor, and the number of tokens is as follows: $H = \lceil \log_{b_f}(N) \rceil$. In our results, we showed the performance by using a binary tree ($b_f = 2$) since this corresponds to a tree design that uses few tokens. By selecting a higher branching factor, the runtime can be decreased at the expense of requiring more tokens (see Table 11). This feature is not available in standard cross attention. As such, TCA enables more control over the memory and computation time trade-off.

## B.2 Additional Experiments

### B.2.1 Copy Task

| Method | $N = 128$ | |
|---|---|---|
| | % Tokens | Accuracy |
| Cross Attention | 100.0% | **100.0 ± 0.0** |
| Random | — | 8.3 ± 0.0 |
| Perceiver IO | **10.9%** | 17.8 ± 0.0 |
| Perceiver IO | **100.0%** | 100.0 ± 0.0 |
| TCA (Acc) | **10.9%** | **100.0 ± 0.0** |
| TCA (CE) | **10.9%** | **100.0 ± 0.0** |

Table 12: Copy Task Results with accuracy (higher is better) and % tokens (lower is better) metrics.

**Results.** In Table 12, we include additional results for the Copy Task where $N = 128$. Similar to previous results, we see that Cross Attention and TCA are able to solve this task perfectly. In contrast, Perceiver IO is not able to solve this task for the same number of tokens. We also see both TCA using accuracy and TCA using negative cross entropy as the reward were able to solve the task perfectly.

| Method | % Tokens | RBF | Matern 5/2 |
|---|---|---|---|
| CNP (Garnelo et al., 2018a) | — | 0.26 ± 0.02 | 0.04 ± 0.02 |
| CANP (Kim et al., 2019) | — | 0.79 ± 0.00 | 0.62 ± 0.00 |
| NP (Garnelo et al., 2018b) | — | 0.27 ± 0.01 | 0.07 ± 0.01 |
| ANP (Kim et al., 2019) | — | 0.81 ± 0.00 | 0.63 ± 0.00 |
| BNP (Lee et al., 2020) | — | 0.38 ± 0.02 | 0.18 ± 0.02 |
| BANP (Lee et al., 2020) | — | 0.82 ± 0.01 | 0.66 ± 0.00 |
| TNP-D (Nguyen & Grover, 2022) | — | $\mathbf{1.39 \pm 0.00}$ | $\mathbf{0.95 \pm 0.01}$ |
| LBANP (Feng et al., 2023a) | — | 1.27 ± 0.02 | 0.85 ± 0.02 |
| CMANP (Feng et al., 2023b) | — | 1.24 ± 0.02 | 0.80 ± 0.01 |
| Perceiver IO ($L = 4$) | 8.5% | 1.02 ± 0.03 | 0.56 ± 0.03 |
| Perceiver IO ($L = 8$) | 17.0% | 1.13 ± 0.03 | 0.65 ± 0.03 |
| Perceiver IO ($L = 16$) | 34.0% | 1.22 ± 0.05 | 0.75 ± 0.06 |
| Perceiver IO ($L = 32$) | 68.0% | 1.25 ± 0.04 | 0.78 ± 0.05 |
| Perceiver IO ($L = 64$) | 136.0% | 1.30 ± 0.03 | 0.85 ± 0.02 |
| Perceiver IO ($L = 128$) | 272.0% | 1.29 ± 0.04 | 0.84 ± 0.04 |
| Transformer + Cross Attention | 100% | $\mathbf{1.35 \pm 0.02}$ | $\mathbf{0.91 \pm 0.02}$ |
| Perceiver IO | 14.9% | 1.06 ± 0.05 | 0.58 ± 0.05 |
| ReTreever | 14.9% | 1.25 ± 0.02 | 0.81 ± 0.02 |

Table 13: 1-D Meta-Regression Experiments with log-likelihood metric (higher is better).

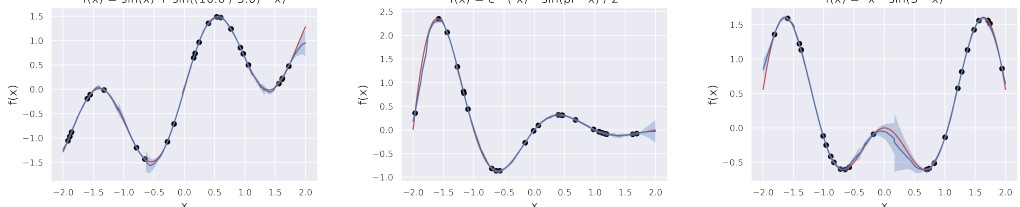

Figure 12: Visualizations of GP Regression

| Method | % Tokens | CelebA |
|---|---|---|
| CNP | — | $2.15 \pm 0.01$ |
| CANP | — | $2.66 \pm 0.01$ |
| NP | — | $2.48 \pm 0.02$ |
| ANP | — | $2.90 \pm 0.00$ |
| BNP | — | $2.76 \pm 0.01$ |
| BANP | — | $3.09 \pm 0.00$ |
| TNP-D | — | $3.89 \pm 0.01$ |
| LBANP (8) | — | $3.50 \pm 0.05$ |
| LBANP (128) | — | $3.97 \pm 0.02$ |
| CMANP | — | $3.93 \pm 0.05$ |
| Perceiver IO ($L = 8$) | 4.1% | $3.18 \pm 0.03$ |
| Perceiver IO ($L = 16$) | 8.1% | $3.35 \pm 0.02$ |
| Perceiver IO ($L = 32$) | 16.2% | $3.50 \pm 0.02$ |
| Perceiver IO ($L = 64$) | 32.5% | $3.61 \pm 0.03$ |
| Perceiver IO ($L = 128$) | 65.0% | $3.74 \pm 0.02$ |
| Transformer + Cross Attention | 100% | $\mathbf{3.88 \pm 0.01}$ |
| Perceiver IO | 4.6% | $3.20 \pm 0.01$ |
| ReTreever | 4.6% | $3.52 \pm 0.01$ |

Table 14: CelebA Image Completion Experiments. The methods are evaluated according to the log-likelihood (higher is better).

### B.2.2    GP Regression

**Results.** In Table 13, we show results for all baselines on GP Regression. We see that Transformer + Cross Attention performs comparable to the state-of-the-art, outperforming all Neural Process baselines except for TNP-D by a significant margin. We evaluated Perceiver IO with varying number of latents $L$. We see that Perceiver IO ($L = 32$) uses $\sim 4.6\times$ the number of tokens to achieve performance comparable to ReTreever. ReTreever by itself already outperforms all NP baselines except for LBANP and TNP-D. Visualizations for different kinds of test functions are included in Figure 12.

### B.2.3    Image Completion (CelebA)

**Results.** In Table 14, we show results for all baselines on CelebA. We see that Transformer + Cross Attention achieves results competitive with the state-of-the-art. We evaluated Perceiver IO with varying numbers of latents $L$. We see that Perceiver IO ($L = 32$) uses $\sim 3.5\times$ the number of tokens to achieve performance comparable to the ReTreever. Visualizations are available in Figure 13.

### B.2.4    Image Completion (EMNIST)

**Results.** In Table 15, we see that Transformer + Cross Attention achieves results competitive with state-of-the-art, outperforming all baselines except TNP-D.

### B.2.5    Time Series: Human Activity

**Results.** In Table 16, we show results on the Human Activity task comparing with several time series baselines. We see that Transformer + Cross Attention outperforms all baselines except for mTAND-Enc. The confidence intervals are, however, very close to overlapping. We would like to note, however, that mTAND-Enc was carefully designed for time series, combining attention, bidirectional RNNs, and reference points. In contrast, Transformer + Cross Attention is a general-purpose model made by just leveraging simple attention modules. We hypothesize the performance could be further improved by leveraging some features of mTAND-Enc, but this is outside the scope of our work.

Notably, ReTreever also outperforms all baselines except for mTAND-Enc. Comparing, Transformer + Cross Attention and ReTreever, we see that ReTreever achieves comparable performance

$N = 25$

$N = 50$

$N = 75$

$N = 100$

$N = 125$

$N = 150$

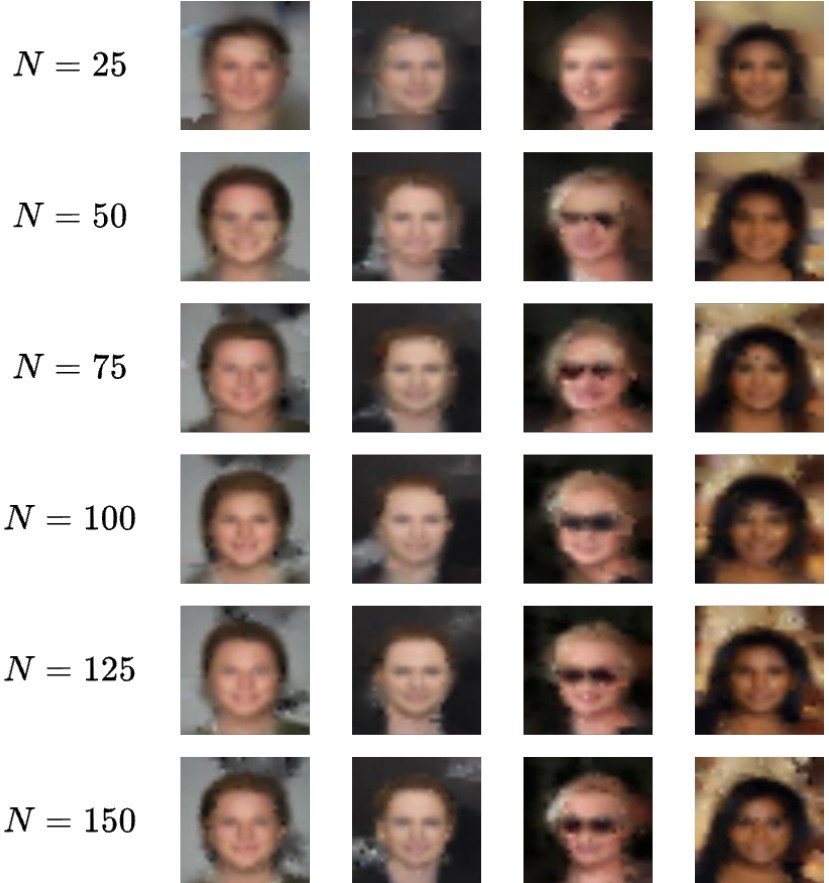

Figure 13: Visualizations of CelebA32

| Method | % Tokens | EMNIST |
|---|---|---|
| CNP | — | 0.73 ± 0.00 |
| CANP | — | 0.94 ± 0.01 |
| NP | — | 0.79 ± 0.01 |
| ANP | — | 0.98 ± 0.00 |
| BNP | — | 0.88 ± 0.01 |
| BANP | — | 1.01 ± 0.00 |
| TNP-D | — | 1.46 ± 0.01 |
| LBANP (8) | — | 1.34 ± 0.01 |
| LBANP (128) | — | 1.39 ± 0.01 |
| CMANP | — | 1.36 ± 0.01 |
| Transformer + Cross Attention | 100% | 1.41 ± 0.00 |
| Perceiver IO | 4.6% | 1.25 ± 0.00 |
| ReTreever | 4.6% | 1.30 ± 0.01 |

Table 15: EMNIST Image Completion Experiments. The methods are evaluated according to the log-likelihood (higher is better).

| Model | % Tokens | Accuracy |
|---|---|---|
| RNN-Impute (Che et al., 2018) | — | 85.9 ± 0.4 |
| RNN-Decay (Che et al., 2018) | — | 86.0 ± 0.5 |
| RNN GRU-D (Che et al., 2018) | — | 86.2 ± 0.5 |
| IP-Nets (Shukla & Marlin, 2019) | — | 86.9 ± 0.7 |
| SeFT (Horn et al., 2020) | — | 81.5 ± 0.2 |
| ODE-RNN (Rubanova et al., 2019) | — | 88.5 ± 0.8 |
| L-ODE-RNN (Chen et al., 2018) | — | 83.8 ± 0.4 |
| L-ODE-ODE (Rubanova et al., 2019) | — | 87.0 ± 2.8 |
| mTAND-Enc (Shukla & Marlin, 2021) | — | 90.7 ± 0.2 |
| Transformer + Cross Attention | 100% | 89.1 ± 1.3 |
| Perceiver IO | 14% | 87.6 ± 0.4 |
| ReTreever | 14% | 88.9 ± 0.4 |

Table 16: Human Activity Experiments with Accuracy metric.

while using far fewer tokens. Furthermore, we see that ReTreever outperforms Perceiver IO significantly while using the same number of tokens.

## C  APPENDIX: IMPLEMENTATION AND HYPERPARAMETERS DETAILS

### C.1  IMPLEMENTATION

For the experiments on uncertainty estimation, we use the official repositories for TNPs (https://github.com/tung-nd/TNP-pytorch) and LBANPs (https://github.com/BorealisAI/latent-bottlenecked-anp). The GP regression and Image Classification (CelebA and EMNIST) datasets are available in the same links. For the Human Activity dataset, we use the official repository for mTAN (Multi-Time Attention Networks) https://github.com/reml-lab/mTAN. The Human Activity dataset is available in the same link. Our Perceiver IO baseline is based on the popular Perceiver (IO) repository (https://github.com/lucidrains/perceiver-pytorch/blob/main/perceiver_pytorch/perceiver_pytorch.py). We report the baseline results listed in Nguyen & Grover (2022) and Shukla & Marlin (2021). When comparing methods such as Perceiver, ReTreever, Transformer + Cross Attention, LBANP, and TNP-D, we use the same number of blocks (i.e., CMABs, iterative attention, and transformer) in the encoder.

### C.2  HYPERPARAMETERS

Following previous works on Neural Processes (LBANPs and TNPs), ReTreever uses 6 layers in the encoder for all experiments. Our aggregator function is a Self Attention (Transformer) module whose output is averaged. Following standard RL practices, at test time, TCA and Re-Treever's policy selects the most confident actions: $u = \text{argmax}_{a \in \mathbb{C}_v} \pi_\theta(a|s)$ We tuned $\lambda_{RL}$ (RL loss term weight) and $\lambda_{CA}$ (CA loss term weight) between $\{0.0, 0.1, 1.0, 10.0\}$ on the Copy Task (see analysis in main paper) with $N = 256$. We also tuned $\alpha$ (entropy bonus weight) between $\{0.0, 0.01, 0.1, 1.0\}$. We found that simply setting $\lambda_{RL} = \lambda_{CA} = 1.0$ and $\alpha = 0.01$ performed well in practice on the Copy Task where $N = 256$. As such, for the purpose of consistency, we set $\lambda_{RL} = \lambda_{CA} = 1.0$ and $\alpha = 0.01$ was set for all tasks. We used an ADAM optimizer with a standard learning rate of $5e - 4$. All experiments were run with 3 seeds. Cross Attention and Tree Cross Attention tended to get stuck in a local optimum on the Copy Task, so to prevent this, dropout was set to 0.1 for the Copy Task for all methods. For image data, we selected a single axis to split the data according to the k-d tree. In the case of EMNIST, the data was split according to the first axis. In the case of CelebA, the data was split according to the second axis.

### C.3  COMPUTE

Our experiments were run using a mix of Nvidia GTX 1080 Ti (12 GB) or Nvidia Tesla P100 (16 GB) GPUs. GP regression experiments took $\sim 2.5$ hours. EMNIST experiments took $\sim 1.75$ hours. CelebA experiments took $\sim 15$ hours. Human Activity experiments took $\sim 0.75$ hours.

