# OpenReview forum: "Tree Cross Attention"
_ICLR.cc/2024/Conference — ICLR 2024 poster_

### Official Review · Reviewer_rpWY · 2023-10-25

**Soundness:** 3 good
**Presentation:** 3 good
**Contribution:** 3 good
**Rating:** 5
**Confidence:** 2

**Summary:**

1. The authors introduce a new module called Tree Cross Attention (TCA) that reduces the number of tokens required for efficient inference while maintaining comparable performance to Cross Attention.

TCA organizes data in a tree structure and performs a tree search at inference time to retrieve relevant tokens for prediction. Specifically, TCA only retrieves information from a logarithmic O(log(N)) number of tokens for performing inference, while Cross Attention scans the full set of O(N) tokens.

2. The authors also present ReTreever, a flexible architecture for token-efficient inference that incorporates TCA.

3. Empirically, the paper demonstrates the effectiveness of TCA and ReTreever on various classification and uncertainty regression tasks.

**Strengths:**

1. The paper is well-written and straightforward to understand.

2. The empirical results that the authors show are impressive. Specifically, the authors show that Perceiver IO's performance drops significantly as the length of the sequence increases, while ReTreever is able to maintain high accuracy across a range of sequence lengths.

3. The proposed method, ReTreever, is able to perform token-efficient inference while achieving better performance than Perceiver IO for the same number of tokens. ReTreever does this by using Tree Cross Attention to retrieve the necessary tokens, only needing a logarithmic number of tokens log(N) << N, making it efficient regardless of the encoder used.

**Weaknesses:**

1. In the evaluation, the authors focus on %tokens metric. How does that translate to wall clock speed up? Or does the tree structure introduce operations that are hard to take advantage of by modern hardware?

**Questions:**

1. My main question regarding the evaluation is in the weakness section.

---

> ### Author Response · Authors · 2023-11-19
> **Response to Reviewer rpWY**
>
> We would like to thank the reviewer for the comments and helpful feedback.
>
> > In the evaluation, the authors focus on % tokens metric. How does that translate to wall clock speed up?
>
> The % tokens do not directly translate to wall clock time. In this work, we focused on % tokens as the metric since it is (1) neither hardware nor implementation specific and (2) % tokens translates towards memory which is a major limiting factor scalability, particularly for (i) attention-based models which are memory intensive and (ii) low-memory devices which are popular (e.g., IoT devices). In contrast, wall clock time (runtime) is highly dependent on the efficiency of the implementation and the hardware. Nonetheless, in the table below, we compare the wall-clock time of the modules used during inference by ReTreever, Transformer+Cross Attention, and Perceiver IO, i.e., Tree Cross Attention, Cross Attention, and Perceiver IO's CA respectively.
>
> We evaluate the modules on the Copy Task for both a GPU and CPU. Although Tree Cross Attention is slower, all methods only require milliseconds to perform inference. Both Tree Cross Attention and Cross Attention learn to solve the task perfectly. However, Tree Cross Attention uses only $3.5$% of the tokens. As such, Tree Cross Attention trades off between the number of tokens and computational time. Perceiver IO, however, fails to solve the task for the same number of tokens.
>
> |         Model        | % Tokens |     Accuracy    | CPU time (in Milliseconds) | GPU Time (in Milliseconds) |
> |:--------------------:|:--------:|:---------------:|:--------------------------:|:--------------------------:|
> |    Cross Attention   |  100.0%  | **100.0 ± 0.0** |            12.05           |            1.61            |
> | Tree Cross Attention | **3.5%** | **100.0 ± 0.0** |            19.31           |            9.09            |
> |   Perceiver IO's CA  | **3.5%** |    13.4 ± 0.2   |          **10.98**         |          **1.51**          |
>
>
>
> > Or does the tree structure introduce operations that are hard to take advantage of by modern hardware?
>
>
> The tree approach has a sequential aspect to it related to the height. However, the major aspects of it can still be parallelized on GPUs. For example, during the tree construction, left and right subtrees can be constructed at the same time since they do not share children nodes. Aggregation can also be performed at the same time for nodes at the same depth. Our implementation leverages parallelization in some aspects of the tree approach, but we have not utilized full-parallelization which can lead to further code optimization and runtime efficiency.
>
> We will be sharing our implementation alongside the camera-ready. We hope it will help improve the efficiency of future implementations of ReTreever and Tree Cross Attention.

---

### Official Review · Reviewer_Wmtm · 2023-10-29

**Soundness:** 3 good
**Presentation:** 2 fair
**Contribution:** 3 good
**Rating:** 5
**Confidence:** 3

**Summary:**

This work proposes a tree cross attention (TCA) architecture to effectively encode context into a tree structure when performing attention, such that the amount of attended tokens will be less than the actual number of context tokens. The architecture is co-trained with the objective to effectively retrieve context through graph-based searching, and utilize context for performing the task. Experiments on a specific set of tasks (i.e., copy task, gp regression on image completion and time series on human activities) show that TCA is able to achieve a similar performance as full cross attention while attending to much fewer tokens.

The architecture proposed is generally applicable to a lot of settings broadly, yet the evaluation is too specific and does not adequately proves the generality of the method.

**Strengths:**

The idea to construct context as a tree is interesting and could have broad implications in constructing context for language models use cases including agent trajectories, in-context learning examples and retrieved documents and more.

**Weaknesses:**

- **More context on the baseline IO Perceiver**: The authors need a background section for IO perceiver so the work is self-contained. With the current version, IO perceiver, though being a famous and well-cited paper, is not clearly stated.
- **Speedup by attending to fewer context tokens**: One claimed benefit of the method is that it attends to fewer tokens to context when performing the task, which I assume would result in an inference speedup. But the work does not explicitly measure if TCA runs faster than CA. On the other hand, TCA does in most of the cases leads to a slight performance degradation, and it will be important to justify the design with proper inference wall-clock time measurements.
- **More general evaluation**: The authors claim to have proposed this general architecture, but the evaluating tasks are specific and not as general as expected. The tasks evaluated in the baseline work — IO perceiver (e.g., pretraining MLM, optical flow and multimodal encoding) seem harder and more general than the ones performed in this paper, (i.e., copy task, gp regression on image completion and time series on human activities). It would be nice to see experimental setups with more significant implications like pre-training, yet there seems to be significant amount of work to be put into actually scale it to these real settings.

**Questions:**

- Does the algorithm run faster than full attention in terms of wall-clock time?

---

> ### Author Response · Authors · 2023-11-19
> **Response to Reviewer Wmtm (1/2)**
>
> We would like to thank the reviewer for the comments and helpful feedback.
>
> > More context on the baseline IO Perceiver: The authors need a background section for IO perceiver so the work is self-contained. With the current version, IO perceiver, though being a famous and well-cited paper, is not clearly stated.
>
> Thank you for bringing this to our attention. We have added a background section for Perceiver IO.
>
> > More general evaluation: [...] The tasks evaluated in the baseline work — IO perceiver (e.g., pretraining MLM, optical flow and multimodal encoding) seem harder and more general than the ones performed in this paper, (i.e., copy task, gp regression on image completion and time series on human activities). It would be nice to see experimental setups with more significant implications like pre-training, yet there seems to be a significant amount of work to be put into actually scaling it to these real settings.
>
> We initially considered evaluating on Perceiver IO's tasks. However, Perceiver IO's experiments used a very large amount of computing resources for their experiments which was significantly outside our capabilities. For example, their experiments used "a batch size of 1024 and 64 TPUs"  (Appendix A.1 in Perceiver IO's paper), "a larger batch size of 8192 and train on 256 TPUs" (Appendix A.2 in Perceiver IO's paper), and "1024 batch size on 64 TPUs"  (Appendix A.2 in Perceiver IO's paper). Specifically for their pre-training experiments in Perceiver IO, the authors used 256 TPUs. In contrast, each of our experiments was run on 1 GPU at a time (Nvidia P100 GPU (16 GB)).
>
> To ensure our experiments were general and with significant implications, we considered (1) different forms of data such as images and time series and (2) popular and well-benchmarked tasks within their respective literatures:  Neural Processes (GP Regression and Image Completion) and Time Series (Human Activity). Time Series models are popular in various important fields such as healthcare, climate science, ecology, and astronomy. Similarly, Neural Processes have been applied to a wide array of impactful settings (see the recent survey (Jha et al., 2023) for a full list) such as climate modelling and detecting biomarkers in fMRI scans.
>
> The datasets we considered are the standard benchmark datasets for evaluating the performance of Neural Processes (GP regression and image completion) and Time Series (Human Activity) models. Our results are competitive with that of state-of-the-art while requiring only a logarithmic number of tokens. Due to space limitations, these comprisons were included in the Appendix. Specifically, the tables in the Appendix include results comparing the performance of ReTreever with a total of $18$ baselines: $9$ baselines for Neural Processes and $9$ baselines for Time Series.
>
>
> ---
>
> Saurav Jha, Dong Gong, Xuesong Wang, Richard E. Turner, Lina Yao. "The neural process family: Survey, applications and perspectives." arXiv preprint arXiv:2209.00517, 2023.

---

> ### Author Response · Authors · 2023-11-19
> **Response to Reviewer Wmtm (2/2)**
>
> > One claimed benefit of the method is that it attends to fewer tokens to context when performing the task, which I assume would result in an inference speedup.
> > [...] it will be important to justify the design with proper inference wall-clock time measurements. [...]
> > Questions:
> > Does the algorithm run faster than full attention in terms of wall-clock time?
>
>
> In this work, we focused on % tokens as the metric since it is (1) neither hardware nor implementation specific and (2) % tokens translates towards memory which is a major limiting factor scalability, particularly for (i) attention-based models which are memory intensive and (ii) low-memory devices which are popular (e.g., IoT devices). In contrast, wall clock time (runtime) is highly dependent on the efficiency of the implementation and the hardware. Nonetheless, in the table below, we compare the wall-clock time of the attention mechanisms.
>
>
> We evaluate the modules on the Copy Task for both a GPU and CPU. Although Tree Cross Attention is slower, all methods only require milliseconds to perform inference. Both Tree Cross Attention and Cross Attention learn to solve the task perfectly. However, Tree Cross Attention uses only $3.5$% of the tokens. As such, Tree Cross Attention trades off between the number of tokens and computational time. Due to space limitations, we have added these results to the appendix.
>
> We will be sharing our implementation alongside the camera-ready. We hope it will help improve the efficiency of future implementations of ReTreever and Tree Cross Attention.
>
> |         Model        | % Tokens |     Accuracy    | CPU time (in Milliseconds) | GPU Time (in Milliseconds) |
> |:--------------------:|:--------:|:---------------:|:--------------------------:|:--------------------------:|
> |    Cross Attention   |  100.0%  | 100.0 ± 0.0 |            12.05           |            1.61            |
> | Tree Cross Attention | 3.5% | 100.0 ± 0.0 |            19.31           |            9.09            |

---

> > ### Comment · Reviewer_Wmtm · 2023-11-20
> > **Response**
> >
> > Thank you for your response! I understand the reasoning behind testing the algorithm in a more small-scale setting, particularly with limited compute resources. However, my concerns about the practical applicability of the method remain given its extreme high cost to possibly to replace cross attention, therefore I will maintain my current score.

---

> ### Author Response · Authors · 2023-11-20
> **Response to Reviewer Wmtm's Response**
>
> Thank you for your prompt response!
>
> > my concerns about the practical applicability of the method remain given its extreme high cost to possibly to replace cross attention, therefore I will maintain my current score.
>
> We would like to reiterate that Tree Cross Attention (TCA) trades off the memory and runtime. Although the runtime increases with TCA, the memory usage (and % tokens) has decreased significantly as shown in Figure 4 and the table results.
>
> The trade-off between memory and runtime can be controlled by the design of the tree. Tree Cross Attention sequentially searches down a tree for the relevant tokens. As such, the factor which primarily affects the overall runtime is the height of tree ($H$) which is dependent on the branching factor $b_f \geq 2$. The relationship between the height of the tree, the branching factor, and the number of tokens is as follows: $H = \lceil\log_{b_f}(N)\rceil$. In our results, we showed the performance with using a binary tree ($b_f=2$). Notably, this corresponds to a tree design with a longer runtime but uses few tokens. By selecting a higher branching factor, the runtime can be decreased at the expense of requiring more tokens as we show in the table below. This feature is not available in standard cross attention.
>
> From these results, it is clear that TCA enables more control over the memory and computation time trade-off. Thus making it more tailored to many practical use cases (especially the low memory settings)
>
> | Model                              | Tree Height ($H$) | % Tokens  | CPU Time (in Milliseconds) | GPU Time (in Milliseconds) |
> |------------------------------------|-------------------|-----------|----------------------------|----------------------------|
> | Cross Attention                    | N/A               | $100.0$ % | 12.05                      | 1.61                       |
> | Tree Cross Attention ($b_f = 256$) | 1                 | $100.0$ % | 16.21                      | 2.05                       |
> | Tree Cross Attention ($b_f = 32$)  | 2                 | $15.2$ %  | 14.05                      | 3.99                       |
> | Tree Cross Attention ($b_f = 16$)  | 2                 | $12.1$ %  | 13.25                      | 4.13                       |
> | Tree Cross Attention ($b_f = 8$)   | 3                 | $7.0$ %   | 15.53                      | 5.73                       |
> | Tree Cross Attention ($b_f = 4$)   | 3                 | $3.9$ %   | 16.30                      | 6.45                       |
> | Tree Cross Attention ($b_f = 2$)   | 8                 | $3.5$ %   | 21.92                      | 9.83                       |

---

### Official Review · Reviewer_779Z · 2023-11-03

**Soundness:** 3 good
**Presentation:** 3 good
**Contribution:** 3 good
**Rating:** 8
**Confidence:** 3

**Summary:**

This paper presents a method for reducing the test-time computational cost of attention. Namely, TCA walks down a tree to attend to a single set of sibling leaves, while only attending to a compressed version of the other leaves. This allows the complexity of attention to be logarithmic in the total number of leaves $N$ (the sequence length). This comes at the cost of training a policy to traverse the tree, which must be trained via REINFORCE, an aggregator that compresses and composes leaf representations, as well as defining the tree itself.

Experiments on a copy task, GP regression, image completion, human activity classification show that the method is efficient and performant. Additional analysis highlights the method's memory efficiency compared to full attention.

**Strengths:**

I enjoyed reading the paper. The paper is well-written and easy to follow. The idea is simple and clever.

**Weaknesses:**

Overall, I believe the paper is pretty complete. I am mostly curious about how to make this method work for self-attention and (masked) autoregressive modeling.

Larger-scale experiments would be appreciated, as the current experiments are quite small-scale. Presumedly the challenges of training the tree expansion policy would increase with harder datasets.

One suggestion for a larger-scale experiment would be training a translation or summarization model and replacing the encoder attention with tree cross attention.

**Questions:**

## Questions
1. Would TCA work out of the box for masked language modeling, e.g. BERT?
2. Did you try using Gumbel-softmax for training the tree expansion policy?
3. What are the barriers to applying TCA to self-attention? Would aggregation become the most expensive operation?

## Suggestions
1. In the last paragraph of 3.1, I was a little confused about why k-d trees were needed as I was only thinking about 1D sequences. Having a picture of an image tree and some more prose about different domains would be really nice for motivating and showing the generality of the method.
2. While including cross attention in the model name pragmatically implies the method is not intended directly for self-attention, it would be nice to add a footnote that the focus is not on autoregressive modeling.

---

> ### Author Response · Authors · 2023-11-19
> **Response to Reviewer 779Z**
>
> We would like to thank the reviewer for their very supportive feedback. We are pleased to see your enthusiasm for our work.
>
>
> > Would TCA work out of the box for masked language modeling, e.g. BERT?
>
> TCA supports masked inputs and can be applied similarly to self-attention by setting the query vectors in TCA the same as the context vectors. As such, TCA can be applied to masked language modelling.
>
> > Did you try using Gumbel-softmax for training the tree expansion policy?
>
> We have not tried using Gumbel-softmax for training the expansion policy but that would be an interesting direction to explore as an alternative to our REINFORCE-based policy training method.
>
>
> > What are the barriers to applying TCA to self-attention? Would aggregation become the most expensive operation?
>
> TCA can be applied to self-attention by setting the query vectors the same as the context vectors. Empirically, we found that aggregation to be a rather efficient operation. For example, on a batch of sequences of length $256$, it took 0.012864 ± 0.002792 seconds.
>
>
>
> > In the last paragraph of 3.1, I was a little confused about why k-d trees were needed as I was only thinking about 1D sequences. Having a picture of an image tree and some more prose about different domains would be really nice for motivating and showing the generality of the method.
>
> Thank you for letting us know! We have added in Section 3.1 more details regarding the construction of k-d trees in different domains (sequences and images) to improve the clarity.
>
> > While including cross attention in the model name pragmatically implies the method is not intended directly for self-attention, it would be nice to add a footnote that the focus is not on autoregressive modelling.
>
> Thank you for your suggestion. We have updated the paper to include a footnote that clarifies the focus is not on autoregressive modelling.

---

### Official Review · Reviewer_RnYk · 2023-11-06

**Soundness:** 3 good
**Presentation:** 4 excellent
**Contribution:** 4 excellent
**Rating:** 8
**Confidence:** 4

**Summary:**

This paper proposes TCA, a tree-based cross attention module to reduce the complexity of cross attention from $O(N)$ to $O(\log(N))$, where $N$ is the number of tokens used for cross attention. Given $N$ tokens, TCA first constructs a balanced binary tree representation using standard methods like K-D tree, where the leaf nodes are the token embeddings, and the internal node representations are aggregated using the two children of the internal node. TCA uses reinforcement learning to learn good internal node representations. This construction is only performed once for a set of context tokens. Now, for a given query vector, a tree search is performed to select a subset of nodes ($O(\log(N))$ size) of the tree for cross attention, resulting in $O(\log(N))$ complexity for retrieval. Using TCA, the paper further proposes ReTreever, a general-purpose retrieval model that achieves token-efficient inference. The paper compares the ReTreever models with other token-efficient retrieval models like Perceiver IO and show impressive gains over the baseline - little to no drop in performance while leveraging only a small subset of tokens for cross attention.

**Strengths:**

- The paper is well written and builds the theory coherently.
- The proposed cross-attention architecture, TCA, along with the general purpose retrieval model, ReTreever is novel.
- Because ReTreever uses reinforcement learning to learn the internal node representations, the reward used for optimization can be non-differentiable like accuracy, which improves performance over a reward based on cross entropy because the reward model is simpler in case of accuracy.
- The reasoning behind each of the loss terms in $\mathcal{L_{ReTreever}}$ is well-explained and it also uses leaf-level cross attention loss to make the training faster.
- The empirical results on various tasks like copying, uncertainty estimation are impressive using ReTreever, and the paper also has good ablation studies to test the various components of the proposed approach.

**Weaknesses:**

- It would be good if a similar row (as given in Table 2) can be added to Table 1 for Perceiver IO with increased latent tokens that matches the performance of TCA on the copy task.
- Theoretical complexity is fine, but the paper should also report wall-clock time for ReTreever and compare it with the full Transformer+Cross Attention and Perceiver IO models. I am guessing the tree approach is not parallelizable on accelerated devices like GPUs, but it would be good to see if there's considerable decrease in latency on CPUs.
- Building on the previous point, wall-clock times for the tree construction and bottom-up aggregation should be reported too.
- Using ReTreever-full does not make sense and it only confuses the understanding of the reader in my opinion. Either remove it, or add more details like why there is a performance gap between the full cross-attention and ReTreever-full given both are using 100% of the tokens.

**Questions:**

I have asked most of my questions in the weakness section. If the authors can address my questions and add the relevant latency benchmarks too, I am willing to increase my score.

---

> ### Author Response · Authors · 2023-11-19
> **Response to Reviewer RnYk (1/2)**
>
> We would like to thank the reviewer for their very constructive feedback and for their support.
>
>
> > It would be good if a similar row (as given in Table 2) can be added to Table 1 for Perceiver IO with increased latent tokens that matches the performance of TCA on the copy task.
>
> In the following table, we include results evaluating Perceiver IO with increased number of latent tokens on varying lengths of the copy task ($N \in \{128, 256, 512, 1024\}$). Specifically, we tested using $100\%$ tokens to evaluate its performance. Although Perceiver IO was able to solve the copy task for $N=128$. Perceiver's performance degraded sharply as the difficulty of the task increased even with $100\%$ tokens. Furthermore, due to Perceiver IO's encoder which aims to map the context tokens to a set of latent tokens, we ran out of memory (on a Nvidia P100 GPU (16 GB)) trying to train the model on a sequence length of 1024.
>
> |                      | $N=128$  | $N=128$     | $N=256$  | $N=256$     | $N=512$  | $N=512$     | $N=1024$ | $N=1024$   |
> |----------------------|----------|-------------|----------|-------------|----------|-------------|----------|------------|
> | Method               | % Tokens | Accuracy    | % Tokens | Accuracy    | % Tokens | Accuracy    | % Tokens | Accuracy   |
> | Cross Attention      | 100.0%   | 100.0 ± 0.0 | 100.0%   | 100.0 ± 0.0 | 100.0%   | 100.0 ± 0.0 | 100.0%   | 99.9 ± 0.2 |
> | Random               | ---      | 8.3 ± 0.0   | ---      | 8.3 ± 0.0   | ---      | 8.3 ± 0.0   | ---      | 8.3 ± 0.0  |
> | Tree Cross Attention | 10.9%    | 100.0 ± 0.0 | 6.3%     | 100.0 ± 0.0 | 3.5%     | 100.0 ± 0.0 | 2.0%     | 99.6 ± 0.6 |
> | Perceiver IO         | 10.9%    | 17.8 ± 0.0  | 6.3%     | 15.2 ± 0.0  | 3.5%     | 13.4 ± 0.2  | 2.0%     | 11.6 ± 0.4 |
> | Perceiver IO         | 100.0%   | 100.0 ± 0.0 | 100.0%   | 63.6 ± 45.4 | 100.0%   | 14.1 ± 2.3  | 100.0%   | OOM        |
>
>
> To analyze the reason behind the degradation in performance of Perceiver IO, we evaluated Perceiver IO on the copy task with a varying number of latent tokens (see table below), i.e., $L \in \{16,32,64,128\}$ and varying sequence lengths $N \in \{128,256,512\}$.  Generally, we found that performance improved as the number of latents increased.  However, the results were relatively unstable. For some runs, Perceiver IO was able to solve the task completely. Other times, Perceiver IO got stuck in lower accuracy local optimas. For longer sequences $N=512$, increasing the number of latents did not make a difference as the model simply got stuck in poor local optimas.
>
> We hypothesize that the poor performance is due to the incompatibility between the nature of the Copy Task and Perceiver IO's model. The copy task (1) has high intrinsic dimensionality that scales with the number of tokens and (2) does not require computing higher-order information between the tokens. In contrast, Perceiver IO (1) is designed for tasks with lower intrinsic dimensionality by compressing information via its latent tokens and iterative attention encoder, and (2) computes higher-order information via a transformer in the latent space. Naturally, these factors make learning the task difficult for Perceiver IO. Due to space limitations, we have added these tables and analyses to the appendix.
>
> | Perceiver IO's Num. Latents ($L$) | Copy Task ($N=128$) Accuracy | Copy Task ($N=256$) Accuracy | Copy Task($N=512$) |
> |:---------------------------:|:-------------------:|:-------------------:|:------------------:|
> |            $L=16$           |    39.83 ± 40.21    |     15.16 ± 0.03    |    13.41 ± 0.15    |
> |            $L=32$           |    58.92 ± 47.44    |    34.75 ± 39.14    |    19.41 ± 12.07   |
> |            $L=64$           |    79.45 ± 41.11    |    27.10 ± 21.23    |    14.00 ± 0.72    |
> |           $L=128$           |    100.00 ± 0.00    |    63.50 ± 39.37    |    13.23 ± 0.42    |

---

> ### Author Response · Authors · 2023-11-19
> **Response to Reviewer RnYk (2/2)**
>
> > Theoretical complexity is fine, but the paper should also report wall-clock time for ReTreever and compare it with the full Transformer+Cross Attention and Perceiver IO models.
>
> In this work, we focused on % tokens as the metric since it is (1) neither hardware nor implementation specific and (2) % tokens translates towards memory which is a major limiting factor scalability, particularly for (i) attention-based models which are memory intensive and (ii) low-memory devices which are popular (e.g., IoT devices). In contrast, wall clock time (runtime) is highly dependent on the efficiency of the implementation and the hardware. Nonetheless, in the table below, we compare the wall-clock time of the modules used during inference by ReTreever, Transformer+Cross Attention, and Perceiver IO, i.e., Tree Cross Attention, Cross Attention, and Perceiver IO's CA respectively.
>
>
>
> We evaluate the modules on the Copy Task for both a GPU and CPU. Although Tree Cross Attention is slower, all methods only require milliseconds to perform inference. Both Tree Cross Attention and Cross Attention learn to solve the task perfectly. However, Tree Cross Attention uses only $3.5$% of the tokens.  As such, Tree Cross Attention trades off between the number of tokens and computational time. Perceiver IO, however, fails to solve the task for the same number of tokens.  Due to space limitations, we have added these results to the appendix.
>
>
> |         Model        | % Tokens |     Accuracy    | CPU time (in Milliseconds) | GPU Time (in Milliseconds) |
> |:--------------------:|:--------:|:---------------:|:--------------------------:|:--------------------------:|
> |    Cross Attention   |  100.0%  | **100.0 ± 0.0** |            12.05           |            1.61            |
> | Tree Cross Attention | **3.5%** | **100.0 ± 0.0** |            19.31           |            9.09            |
> |   Perceiver IO's CA  | **3.5%** |    13.4 ± 0.2   |          **10.98**         |          **1.51**          |
>
>
>
> > I am guessing the tree approach is not parallelizable on accelerated devices like GPUs
>
> The tree approach has a sequential aspect to it related to the height. However, significant aspects of it can still be parallelized on GPUs. For example, during the tree construction, left and right subtrees can be constructed at the same time since they do not share children nodes. Aggregation can also be performed at the same time for nodes at the same depth. Our implementation leverages parallelization in some aspects of the tree approach, but we have not utilized full-parallelization which can lead to further code optimization and runtime efficiency.
>
> We will be sharing our implementation alongside the camera-ready. We hope it will help improve the efficiency of future implementations of ReTreever and Tree Cross Attention.
>
>
>
> > Building on the previous point, wall-clock times for the tree construction and bottom-up aggregation should be reported too.
>
> For $256$ tokens, the wall-clock time for constructing a tree is $0.076 ± 0.022$ milliseconds. The wall-clock time for the aggregation is $12.864 ± 2.792$ milliseconds. We have added these numbers to the appendix.
>
>
>
> > Using ReTreever-full does not make sense and it only confuses the understanding of the reader in my opinion. Either remove it, or add more details like why there is a performance gap between the full cross-attention and ReTreever-full given both are using 100% of the tokens.
>
> Thank you for your suggestion! We have removed ReTreever-Full from the main paper as it is not the focus.

---

> > ### Comment · Reviewer_RnYk · 2023-11-22
> > **Response to Rebuttal**
> >
> > Thank you for additional experimental results and the detailed response. Although I am a bit skeptical about the wall clock time compared to the baseline, I believe the paper proposes a novel tree-based approach for cross attention using a small subset of tokens. Therefore, I am recommending this paper for acceptance and increasing my rating to 8.

---

### Author Response · Authors · 2023-11-22
**Message to Reviewers**

Dear Reviewers,

We appreciate your time and consideration. We have responded to all your comments below. We are happy to address any further questions you have. Any feedback would be highly appreciated. We look forward to hearing from you.

---

### Meta-Review · Area_Chair_WGbc · 2023-12-13

**Metareview:**

**Paper Summary:**

This paper introduces Tree Cross Attention (TCA), a method that reduces the complexity of cross-attention. TCA achieves this by organizing data in a tree structure, allowing for information retrieval from a logarithmic number of tokens. Building upon TCA, the authors develop ReTreever, a model which demonstrates superior performance compared to Perceiver IO when utilizing the same number of tokens.

**Strengths:**

1. Novelty: TCA is offers a novel solution to the challenge of token efficiency in attention mechanisms (RnYk, Wmtm).
2. Strong Empirical Results: The paper presents strong results across various tasks, showing particular effectiveness as sequence length increases (RnYk, rpWY).

**Weaknesses:**

1. Higher Inference Latency: TCA exhibits increased inference latency compared to Perceiver IO, especially on GPUs. While TCA allows for a trade-off between wall-clock time and accuracy, Perceiver IO also allows the trade-off by varying the number of latents used. It is recommended that the authors include a detailed trade-off analysis between wall-clock time and accuracy in future versions (Wmtm, rpWY, Response to Reviewer RnYk (2/2)).
2. Need for Larger-Scale Experiments: The evaluation of TCA predominantly involves small-scale experiments. Its effectiveness in more complex, real-world tasks such as translation, summarization, or pretraining remains uncertain. Showing the feasibility of learning the tree expansion policy in these larger-scale applications would greatly enhance the paper's impact (779Z, Wmtm).

**Final Decision:**

Despite concerns about the method's performance on more challenging tasks and the absence of a comprehensive analysis comparing wall-clock time and accuracy to existing baselines, considering the overall strengths of the paper, particularly its novelty and strong results, I recommend accepting this paper.

**Justification For Why Not Higher Score:**

The evaluation of TCA predominantly involves small-scale experiments. Its effectiveness in more complex, real-world tasks remains uncertain.

**Justification For Why Not Lower Score:**

TCA is a novel method for improving token efficiency and shows strong results on the tasks evaluated in the paper.

---

### Decision · Program_Chairs · 2024-01-16

Accept (poster)